# Disentangled Counterfactual Learning for Physical Audiovisual Commonsense Reasoning

**Changsheng Lv[1,2∗], Shuai Zhang[1,2∗], Yapeng Tian[3], Mengshi Qi[1,2⊠†], and Huadong Ma[1,2]**

[1]Beijing Key Laboratory of Intelligent Telecommunications Software and Multimedia
[2]Beijing University of Posts and Telecommunications
[3]Department of Computer Science, The University of Texas at Dallas
{lvchangsheng, zshuai, qms, mhd}@bupt.edu.cn, yapeng.tian@utdallas.edu

## Abstract

In this paper, we propose a Disentangled Counterfactual Learning (DCL) approach for physical audiovisual commonsense reasoning. The task aims to infer objects' physics commonsense based on both video and audio input, with the main challenge is how to imitate the reasoning ability of humans. Most of the current methods fail to take full advantage of different characteristics in multi-modal data, and lacking causal reasoning ability in models impedes the progress of implicit physical knowledge inferring. To address these issues, our proposed DCL method decouples videos into static (time-invariant) and dynamic (time-varying) factors in the latent space by the disentangled sequential encoder, which adopts a variational autoencoder (VAE) to maximize the mutual information with a contrastive loss function. Furthermore, we introduce a counterfactual learning module to augment the model's reasoning ability by modeling physical knowledge relationships among different objects under counterfactual intervention. Our proposed method is a plug-and-play module that can be incorporated into any baseline. In experiments, we show that our proposed method improves baseline methods and achieves state-of-the-art performance. Our source code is available at https://github.com/Andy20178/DCL.

## 1 Introduction

Humans acquire physical commonsense knowledge through integrating multi-modal information and have the ability to reason the physical properties of previously unseen objects in the real world [1].

Whether inferring the physical properties of unseen objects ("this object is probably made of wood") or solving practical problems ("which object would make a bigger mess if dropped") [2], such reasoning abilities represent a sophisticated challenge in the realm of machine intelligence, and play the important role in robust robot navigation [3] as well as AR/VR [4]. In this work, we utilize audiovisual question answering as a proxy task to advance the machine's capacity for physical commonsense reasoning.

The major challenge in physical commonsense reasoning is how to develop methods that can extract and reason the implicit physical knowledge from abundant multi-modal data, especially video inputs. This requires the ability to process complex video information, identify the category and the corresponding physical property of each object, and understand the causal relationships among objects. These cognitive processes are similar to those that humans use to know, learn, and reason about the physical world. Current existing methods [5, 6, 7, 8] often extract generic visual features from videos that contain human-object interactions, resulting in a mixed feature representation that

---

∗Equal contribution
†Corresponding author.

37th Conference on Neural Information Processing Systems (NeurIPS 2023).

does not distinguish between object and action information. However, in physical commonsense reasoning, it is critical to clearly figure out the attributes and physical properties of objects.

To mitigate the challenge, our idea is to meticulously disentangle the whole video content into static parts and dynamic ones, which means they remain constant features over time, and the features vary with time, respectively. Another driving force behind our work is to establish relationships of physical knowledge among various objects within both video and audio modalities. We enhance the optimization of the results by taking into account the relevance of multiple samples and incorporating causal learning, by applying such relationships as confounders. The application of counterfactual interventions further bolsters the model's explainability capacity.

In this paper, we propose a new *Disentangled Counterfactual Learning* (DCL) framework for physical audiovisual commonsense reasoning, which explicitly extracts static and dynamic features from video and uses causal learning to mine implicit physical knowledge relationships among various objects. Specifically, we design a Disentangled Sequential Encoder (DSE), which is based on a sequential variational autoencoder to clearly separate the static and dynamic factors of the input videos through self-supervision. Meanwhile, we incorporate the contrastive estimation method to enhance the mutual information (MI) between the input data and two latent factors, while also discouraging the MI between the static and dynamic factors. Furthermore, we propose a novel Counterfactual Learning Module (CLM) to capture physical knowledge relationships from a wide array of data samples and employ counterfactual intervention with causal effect tools during the reasoning process. We further refine the training objectives of the model by maximizing the probability likelihood in DSE and the Total Indirect Effect value in CLM. Extensive experimental results on PACS dataset [2] validate that our proposed DCL approach achieves superior results over previous state-of-the-art methods, demonstrating our model can effectively improve the reasoning ability and enhance the explainability.

Our main contributions can be summarized as follows:

**(1)** We propose a new Disentangled Counterfactual Learning (DCL) approach for physical audio-visual commonsense reasoning, which disentangles video inputs into static and dynamic factors via a Sequential Variational Autoencoder. To the best of our knowledge, we are the first to adopt disentanglement learning into this specific task.

**(2)** We introduce a novel Counterfactual Learning Module to model physical knowledge relationships among various objects, and then apply them as counterfactual interventions to enhance the ability of causal reasoning.

**(3)** Experimental results demonstrate our proposed DCL is a plug-and-play module that can be incorporated into any baseline, and show DCL can achieve 3.2% absolute improvement over the state-of-the-art methods.

## 2 Related Work

**Physical Commonsense Reasoning.** Naive and intuitive physics was first studied in human psychology experiments [9, 10, 11]. Further research has also been conducted in the multi-sensory perception of physical properties [12]. Nowadays, visual and language are adopted to learn and utilize physical commonsense in different tasks [13, 14, 15, 16, 17], which only focus on single-modality knowledge and disregard multi-modal fusion. In embodied audio-visual AI tasks, existing research utilizes multi-modal information to address inverse physics problems [5, 6, 7, 8]. Different from them, we introduce a disentangled counterfactual learning approach to reason the physical commonsense from both video and audio inputs.

**Disentangled Representation Learning.** Disentangled representation learning aims to learn various hidden explanatory factors behind observable data [18], which has already been widely applied in computer vision and natural language processing [19, 20, 21, 22, 23], such as image-to-image translation [19], image generation [20] and text generation [21, 21]. However, such methods are limited in the Visual Question Answering (VQA) task. In this paper, we propose a VAE-based disentangled learning method that disentangles visual information into static and dynamic aspects and improves the interpretability of the model.

**Causal learning.** Traditional VQA datasets suffer from "language prior" [24] or "visual priming bias" [25], leading current methods focus on features' inherent bias of either language or vision and

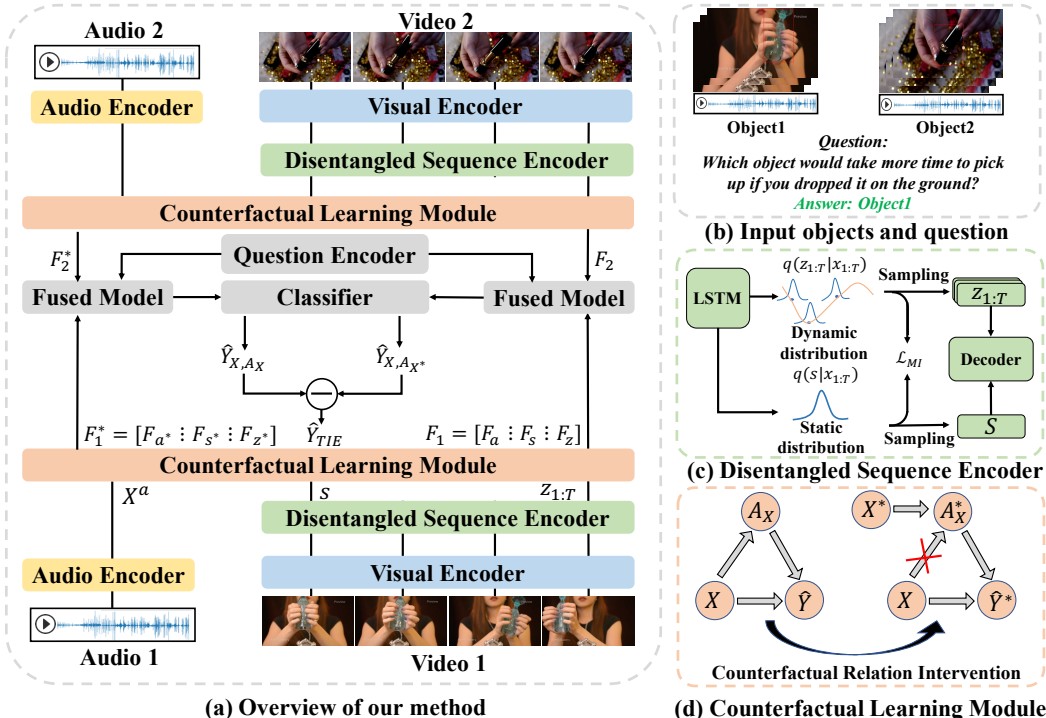

**(a) Overview of our method**

**(b) Input objects and question**

**(c) Disentangled Sequence Encoder**

**(d) Counterfactual Learning Module**

Figure 1: The illustration of our proposed DCL model: (a) shows the overall architecture: the input videos with audios (b) are first encoded by the corresponding visual and audio encoders, then the Disentangled Sequence Encoder (c) is used to decouple video features into static and dynamic factors by LSTM-based VAE. Through the Counterfactual Learning Module (d), we construct the affinity matrix $A$ as a confounder and obtain the prediction $\hat{Y}_{X,A_X}$ and counterfactual result $\hat{Y}_{X,A_X^*}$. Finally, we achieve $\hat{Y}_{TIE}$ by subtracting these two results and optimizing the model by cross-entropy loss.

then output inaccurate answers. In recent years, counterfactual thinking and causal reasoning have been widely applied in visual explanations [26, 27, 28, 29], scene graph generation [30, 31, 32, 33, 34], video understanding [35, 36, 37, 38, 39] and image recognition [31]. Recently, there has been some work on audio-visual video question answering [40, 41, 42], but they generally focus on cross-modal modeling. Our approach, in contrast, focuses on constructing physical knowledge relationships among different samples and using them as confounders in causal reasoning.

## 3 Proposed Approach

### 3.1 Overview

Figure 1(a) illustrates the overall framework of our proposed DCL. First, the model extracts features of input videos and the corresponding audio by visual and audio encoders, respectively. Then, a newly designed Disentangled Sequence Encoder (Sec. 3.2) is used to decouple the video into static and dynamic factors. Subsequently, the raw and intervened multi-modal features are obtained through the Counterfactual Learning Module (Sec. 3.3), and then fused with the question feature. Finally, the prediction result is obtained by optimizing the classification results based on the raw and intervened features (Sec. 3.4).

**Problem Formulation.** As shown in Figure 1(b), given a pair of videos (*i.e.,* $< v_1, v_2 >$) with the corresponding audios (*i.e.,* $< a_1, a_2 >$), where each video contains an object, and our task is to select a more appropriate object from the video inputs to answer the questions (*i.e.,* $q$) about the physical commonsense. We define the estimated answer as $\hat{Y}$ and the ground-truth answer as $Y$ in the dataset. During the pre-processing, we denoted the extracted features of audio, video, and question text as $X^a$, $X^v$, and $X^t$, respectively, where $X^a, X^v, X^t \in \mathbb{R}^d$, $d$ denotes the feature dimension. Especially,

we capture the audio feature and text feature as non-sequence while extracting the video feature as sequence data as $X^v = \{X_1^v, X_2^v, \cdots, X_T^v\}$, where $T$ refers to the number of video frames.

## 3.2 Disentangled Sequential Encoder

The proposed Disentangled Sequential Encoder, as shown in Figure 1(c), can separate the static and dynamic factors for the multi-modal data (we take the video data as the example in the following), which extends the traditional sequential variational autoencoder by incorporating the mutual information term.

Specifically, we assume the latent representations of the input video's feature $X_{1:T}^v$ can divide into the static factor $s$ and the dynamic factors $z_{1:T}$, where $z_t$ is the latent dynamic representation at time step $t$. Following [43, 44], we assume such two factors are independent of each other, i.e., $p(s, z_{1:T}) = p(s)p(z_{1:T})$, where $p(\cdot)$ represents the probability distribution. And $z_i$ depends on $z_{<i} = \{z_0, z_1, .., z_{i-1}\}$, where $z_0 = 0$ and the reconstruction of $x_i$[3] is independent of other frames conditioned on $z_i$ and $s$. Hence we need to learn a posterior distribution $q(z_{1:T}, s|x_{1:T})$ where the two factors are disentangled formulated as:

$$q(z_{1:T}, s|x_{1:T}) = q(z_{1:T}|x_{1:T})q(s|x_{1:T}) = q(s|x_{1:T}) \prod_{i=1}^{T} q(z_i|z_{<i}, x_{\leq i}). \tag{1}$$

We model the posterior distribution by Bi-LSTM [45], where $q(z_i|z_{<i}, x_{\leq i})$ is conditioned on the entire time series by taking the hidden states as the input, and $q(s|x_{1:T})$ can be computed by input $x_{1:T}$. Then we sample the two decoupled factors $s$ and $z_{1:T}$ by utilizing the distribution $q(s|x_{1:T})$ and $q(z_i|z_{<i}, x_{\leq i})$ via the reparameterization trick [46]. Afterward, we use the obtained two disentangled factors to reconstruct the $x_{1:T}$ based on a VAE-based decoder [44]. The prior of static factor $s$ and dynamic factor $z_i$ can be defined as the Gaussian distribution with $\mathcal{N}(0, I)$ and $\mathcal{N}(\mu(z_{<i}), \sigma^2(z_{<i}))$, respectively, where $\mu(\cdot)$ and $\sigma(\cdot)$ are modeled by Bi-LSTM. The reconstruction process can be formalized by the factorization:

$$p(x_{1:T}, s, z_{1:T}) = p(s) \prod_{i=1}^{T} p(z_i|z_{<i})p(x_i|z_i, s). \tag{2}$$

Furthermore, we introduce the mutual information to encourage the disentangled two factors mutually exclusive and incorporate non-parameter contrastive estimation into the standard loss function for learning latent representations [43, 47], which can be formulated as:

$$\mathcal{C}(z_{1:T}) = \mathbb{E}_{p_D} log \frac{\phi(z_{1:T}, x_{1:T}^+)}{\phi(z_{1:T}, x_{1:T}^+) + \sum_{j=1}^{n} \phi(z_{1:T}, x_{1:T}^j)}, \tag{3}$$

where $x^+$ means the video has the same object as "positive example", while $x^j, j = \{1, 2, ..., n\}$ is a set of $n$ "negative" videos that have different objects. In order to alleviate the high dimensionality curse [48, 49], we use $\phi(z_{1:T}, x_{1:T}^+) = exp(sim(z_{1:T}, x_{1:T}^+)/\tau)$, where $sim(\cdot, \cdot)$ means cosine similarity function and $\tau = 0.5$ is a temperature parameter. $\mathcal{C}(s)$ can be calculated in the similar way.

Following [44], we use content augmentation by randomly shuffling the time steps of the video, and performing motion augmentation via the Gaussian blur [49]. The results can be denoted as $\mathcal{C}(z_{1:T}^m)$ and $\mathcal{C}(s^m)$, where $z_{1:T}^m$ and $s^m$ are the augmented data of $z_{1:T}$ and $s$, respectively. The MI term can be formulated as follows:

$$MI(z_{1:T}; x_{1:T}) \approx \frac{1}{2}(\mathcal{C}(z_{1:T}) + \mathcal{C}(z_{1:T}^m)), \qquad MI(s; x_{1:T}) \approx \frac{1}{2}(\mathcal{C}(s) + \mathcal{C}(s^m)). \tag{4}$$

The objective function can be formulated by adding MI terms to the vanilla ELBO as follows:

$$\mathcal{L}_{DSE} = \mathbb{E}_{q(z_{1:T}, s|x_{1:T})}[-\sum_{t=1}^{T} \log p(x_t|s, z_t)] + \gamma \cdot [KL(q(s|x_{1:T})||p(s))$$

$$+ \sum_{t=1}^{T} KL(q(z_t|x_{\leq t})||p(z_t|z_{<t}))] - \alpha \cdot MI(z_{1:T}; x_{1:T}) - \beta \cdot MI(s; x_{1:T}) + \theta \cdot MI(z_{1:T}; s), \tag{5}$$

---

[3]For the sake of simplicity, in the following content, we will use $x_i$ to represent $X_i^v$.

where $\gamma$, $\alpha$, $\beta$ and $\theta$ are hyper-parameters. The full proof can be found in [44].

## 3.3 Counterfactual Learning Module

In this module, we obtain the decoupled static and dynamic factors $s$ and $z_{1:T}$ after performing the Disentangled Sequential Encoder. Then, these factors with the corresponding audio features are sent to construct physical knowledge relationships. Meanwhile, we utilize the counterfactual relation intervention to improve knowledge learning.

**Physical Knowledge Relationship.** Inspired by Knowledge Graph [50, 51], the physical knowledge contained in different samples may have a certain correlation. Therefore, we propose to model such implicit relationships as a graph structure and draw the affinity matrix $A$ to represent this physical knowledge relationship among various objects. Similarly, we build an affinity matrix for audio features or other modal data. With the well-constructed affinity matrix $A$, we can enhance the static, dynamic, and audio features denoted as $X_s^v$, $X_z^v$, and $X^a$ through passing the message and transferring across various samples, formulated as:

$$X = \left[ \begin{array}{c|c|c} X^a & X_s^v & X_z^v \end{array} \right], \qquad A_X = \left[ \begin{array}{c|c|c} A_{X^a} & A_{X_s^v} & A_{X_z^v} \end{array} \right], \qquad F = A_X X^\top, \qquad (6)$$

where $A_X$ represents the augmented matrix composed of three affinity matrices, while $F$ refers to the transferred features, and $\top$ denotes the transpose of a matrix. To achieve $A$, we take $A_{X_s^v}$ as an example: we first compute the similarity matrix based on the static features and filter out the noisy relationships by defining a near neighbor chosen function as $\mathcal{T}(\cdot, k)$ to retain the top-k values in each row, and then adopt the Laplacian matrix $D^{-1}$ of $\mathcal{S}'$ to normalize the total affinities as the following:

$$\mathcal{S}^{i,j} = \exp(\frac{sim(x_i, x_j)}{\tau}), x_i, x_j \in X_s \qquad \mathcal{S}' = \mathcal{T}(\mathcal{S}, k), \qquad A_{X_s} = D^{-1} \cdot \mathcal{S}', \qquad (7)$$

where $sim(\cdot, \cdot)$ is the cosine similarity. The calculation of $A_z^v$ and $A^a$ is similar to $A_s^v$.

**Counterfactual Relation Intervention.** To add additional supervision of the affinities $A_X$, we present to highlight the role of the object's physical knowledge relationship during the optimization. First, we express our method as a Structural Causal Model (SCM) [52, 53] as shown in Figure 1(d), and then introduce causal inference into our method. $\hat{Y}$ represents the final classification output of the model, which is obtained by feeding forward F input into the fusion model and classifier:

$$\hat{Y}_{X,A_X} = CLS(Fusion(F_1, F_2, X^t)), \qquad (8)$$

where $F_1$ and $F_2$ denote the fused feature of input video pair $v_1$ and $v_2$, and $X^t$ denotes the feature of the question text. '$CLS$' and '$Fusion$' represent the classifier and fusion model, respectively. Obviously, the process of deriving the output $\hat{Y}$ from input $X$ can be regarded as two types of effects: one is the direct effect $X \to \hat{Y}$, and the other is an indirect type $X \to A_X \to \hat{Y}$. Our final loss function is to maximum likelihood estimation, which will have an end-to-end effect on both types of effects, resulting in insufficient enhancement of $A$ in the indirect effects path. Thus we leverage the Total Indirect Effect (TIE) [54] here to underline the $A_X$:

$$\hat{Y}_{TIE} = \hat{Y}_{X,A_X} - \mathbb{E}_{X^*}[\hat{Y}_{X,A_{X^*}}], \qquad (9)$$

where $\hat{Y}_{X,A_{X^*}}$ refers to calculating the results by replacing the original affinity $A_X$ to a intervened one $A_{X^*}$, and $X^*$ is the given intervened inputs. Note that $\hat{Y}_{X,A_{X^*}}$ cannot happen in the real world because features $X$ and affinities $A_{X^*}$ come from different $X$ and $X^*$, which is called counterfactual intervention. Therefore, we modify $\hat{Y}_{X,A_X}$ to $\hat{Y}_{X,A_{X^*}}$ is equal to keeping all features fixed but only changing the affinity $A$, which can highlight the critical effect of $A$. We compute the expectation of that effect to get the more stable one, and the intervened input features $X^*$ are sampled by a Gaussian distribution:

$$X^* = X_\sigma \cdot W + X_\mu, \qquad (10)$$

where $W$ is the standard random vector whose dimension is as same as $X$, and the mean $X_\mu$ and standard deviation $X_\sigma$ are both learned by the re-parameterization trick in the end-to-end learning.

### 3.4 Fusion Model and Optimization

As we have stated, our approach is a plug-and-play module that can be directly integrated into various multimodal fusion methods. We take Late fusion [55] as an example to illustrate the structure of fusion models, as well as the optimization function of our DCL. The fusion model can be expressed in the following formula:

$$Fusion(F_1, F_2, X^t) = MLP(Concat(MLP(Concat(F_1, F_2)), X^t)), \tag{11}$$

where $Concat$ represents row-wise concatenation, and $MLP$ represents multi-layer perception.

Finally, our final optimization goal can be expressed as:

$$\mathcal{L}_{total} = \mathcal{L}_{DSE} + \mathcal{L}_{TIE}, \tag{12}$$

where $\mathcal{L}_{DSE}$ denote the optimization function we described in Eq. (5) and $\mathcal{L}_{TIE}$ denotes the cross-entropy loss of $\hat{Y}_{X,A_X}$ in Eq. (9) given the ground-truth $Y$.

## 4 Experiments

### 4.1 Experimental Setup

**Dataset.** The Physical Audiovisual CommonSense Reasoning Dataset (PACS) [2] is a collection of 13,400 question-answer pairs designed for testing physical commonsense reasoning capabilities. The dataset contains 1,377 unique physical commonsense questions spanning various physical properties and is accompanied by 1,526 videos and audio clips downloaded from YouTube. The total number of data points in the PACS dataset is 13,400, and for PACS-Material it is 4,349. Following [2], We divide PACS into 11,044/1,192/1,164 as train/val/test set, each of which contains 1,224/150/152 objects respectively. We partitioned the PACS-Material subset into 3,460/444/445 for train/val/test under the same object distribution as PACS. To ensure a fair evaluation of models, we evaluate our method on both the entire dataset and a subset focused on material-related problems, reporting the results separately for each subset during testing.

**Evaluation Metric.** We follow the prior work [2] and use the accuracy to evaluate both PACS and PACS-material subsets. To evaluate the effectiveness of our static factor via disentanglement learning, we measure the accuracy of the PACS-material subset using only the static factor. In our experiments, we report results by running the experiments five times and taking the average following [2].

**Implementation.** We implement our proposed model with PyTorch on two NVIDIA RTX 3090 GPUs. Specifically, we downsampled each video to $T = 8$ frames during pre-processing and set the feature dimension as $d = 256$. In the Disentangled Sequence Encoder, we used a hidden layer size of 256 for Bi-LSTM. During optimization, we set the batch size as 64, which consisted of 64 video pairs and the corresponding questions. In the Counterfactual Learning Module, $\tau = 2$ and $k = 5$ were used when calculating similarities and constructing the physical knowledge relationships. More details about hyper-parameters analysis can be found in our supplementary.

**Compared Baseline Methods.** To demonstrate the effectiveness of our method across multiple models, we apply it to the following baselines: 1) Late fusion [55], which adopts independent text, image, audio, and video encoders to obtain features, and the unimodal embeddings are concatenated and passed through a linear layer to create multimodal embeddings for prediction. 2) CLIP/AudioCLIP [56, 57], which embed video, text, and audio data into a shared vector space using CLIP and AudioCLIP, and a linear layer is used to create multimodal embeddings for prediction. Notes that CLIP is unable to extract audio features, so we did not use audio in our experiments with CLIP. 3) UNITER [58], which is a pre-trained image and text model for four different image-text tasks and performs well on tasks such as NLVR2 [59]. As for all of the above-mentioned baseline methods, we adopt the reported parameters following their corresponding paper. Note that another state-of-the-art Merlot Reserve [60] is a large-scale multimodal pretraining model that learns joint representations of audio, visual, and language by guiding the new pretraining objectives trained on TPU-v3-512. However, due to the restriction of experimental resources, we are unable to test the performance of DCL with this model and leave it as future work.

| Baseline Model | Accuracy (%) | |
| --- | --- | --- |
| | PACS | PACS-Material |
| Late Fusion [55] | $55.0 \pm 1.1$ | $67.4 \pm 1.5$ |
| Late Fusion w/ DCL | $57.7 \pm 0.9$ | $69.7 \pm 1.2$ |
| CLIP [56] | $56.3 \pm 0.7$ | $72.4 \pm 1.1$ |
| CLIP w/ DCL | $58.4 \pm 0.8$ | $75.4 \pm 1.2$ |
| AudioCLIP [57] | $60.0 \pm 0.9$ | $\underline{75.9 \pm 1.1}$ |
| AudioCLIP w/ DCL | $\mathbf{63.2 \pm 0.8}$ | $\mathbf{76.2 \pm 1.4}$ |
| UNITER(Large) [58] | $60.6 \pm 2.2$ | $75.0 \pm 2.8$ |
| UNITER w/ DCL* | $\underline{62.0 \pm 2.4}$ | $75.7 \pm 2.8$ |

(a) Comparison of our DCL with different baselines.

| Baseline Model | Accuracy (%) | |
| --- | --- | --- |
| | PACS | PACS-Material |
| Late Fusion [55] | $55.0 \pm 1.1$ | $67.4 \pm 1.5$ |
| Late Fusion w/ MLP | $54.9 \pm 0.9$ | $67.7 \pm 1.1$ |
| Late Fusion w/ DSE | $56.2 \pm 0.8$ | $68.5 \pm 1.2$ |
| CLIP[56] | $56.3 \pm 0.7$ | $72.4 \pm 1.1$ |
| CLIP w/ MLP | $56.5 \pm 0.5$ | $72.6 \pm 1.2$ |
| CLIP w/ DSE | $57.0 \pm 0.6$ | $73.2 \pm 1.1$ |
| AudioCLIP[57] | $\underline{60.0 \pm 0.9}$ | $\underline{75.9 \pm 1.1}$ |
| AudioCLIP w/ MLP | $60.3 \pm 0.8$ | $76.2 \pm 1.3$ |
| AudioCLIP w/ DSE | $\mathbf{61.1 \pm 0.8}$ | $\mathbf{76.5 \pm 1.0}$ |

(b) Ablation study of DSE.

| Baseline Model | Accuracy (%) | |
| --- | --- | --- |
| | PACS | PACS-Material |
| Late Fusion [55] w/ DSE | $56.2 \pm 0.8$ | $68.5 \pm 1.2$ |
| Late Fusion w/ DSE,A | $57.0 \pm 1.1$ | $68.9 \pm 1.3$ |
| Late Fusion w/ DSE,A,C | $57.7 \pm 0.9$ | $69.7 \pm 1.2$ |
| CLIP [56] w/ DSE | $57.0 \pm 0.6$ | $73.2 \pm 1.1$ |
| CLIP w/ DSE,A | $57.8 \pm 0.8$ | $74.5 \pm 1.1$ |
| CLIP w/ DSE,A,C | $58.4 \pm 0.8$ | $75.4 \pm 1.2$ |
| AudioCLIP [57] w/ DSE | $61.1 \pm 0.8$ | $\underline{76.0 \pm 1.0}$ |
| AudioCLIP w/ DSE,A | $\underline{61.9 \pm 0.9}$ | $75.8 \pm 1.1$ |
| AudioCLIP w/ DSE,A,C | $\mathbf{63.2 \pm 0.8}$ | $\mathbf{76.2 \pm 1.4}$ |

(c) Ablation study of CLM

| Baseline Model | Accuracy (%) | |
| --- | --- | --- |
| | PACS-Material | $\Delta$ |
| Late Fusion [55] w/ dynamic | $60.4 \pm 1.3$ | - |
| Late Fusion w/ static | $66.0 \pm 0.6$ | 5.6 |
| Late Fusion w/ static, dynamic | $69.7 \pm 1.2$ | 3.7 |
| CLIP [56] w/ dynamic | $63.0 \pm 0.7$ | - |
| CLIP w/ static | $71.1 \pm 0.9$ | 8.1 |
| CLIP w/ static, dynamic | $\underline{75.4 \pm 1.2}$ | 4.3 |
| AudioCLIP [56] w/ dynamic | $67.1 \pm 0.6$ | - |
| AudioCLIP w/ static | $72.5 \pm 1.0$ | 5.4 |
| AudioCLIP w/ static, dynamic | $\mathbf{76.2 \pm 1.4}$ | 3.7 |

(d) Ablation analysis of the static factors.

Table 1: Experimental results and analysis in terms of the accuracy achieved on PACS and PACS-Material. 'DSE' refers to Disentangled Sequential Encoder, while 'CLM' refers to Counterfactual Learning Module. 'A' represents physical knowledge relationship, and 'C' stands for Counterfactual Relation Intervention. $\Delta$ represents the performance improvement compared with the previous row.

## 4.2 Comparison to Baselines

**Quantitative Results.** We report the quantitative performance comparisons using the PACS dataset in Table 1(a). The results show that with the addition of our proposed DCL, both the Late Fusion and UNITER models register absolute improvements of 2.7% and 1.4%, respectively. Similarly, CLIP and AudioCLIP models, which align the image, audio, and text into a shared space, achieve absolute improvements of 2.1% and 3.2%, respectively. These results highlight the strong reasoning and generalization capabilities of our DCL method.

Interestingly, even with the DCL, CLIP's performance trails that of AudioCLIP, emphasizing the critical role of audio information. When comparing CLIP and AudioCLIP enhanced with our DCL, it is evident that the inclusion of audio information leads to an absolute improvement of 4.8%. The same conclusion applies to the PACS-Material subset, indicating that our method, serving as a plug-and-play module, can consistently enhance material reasoning performance and be flexibly incorporated across various models. It is noteworthy that all objects in the test set were absent from the training and validation sets, attesting to the zero-shot reasoning ability of our model.

**Qualitative Results.** In order to provide a comprehensive qualitative perspective on our results, we present the visualized results of different models for the same questions in Figure 2. In Figure 2(a), the two objects exhibit significant visual and auditory differences, facilitating accurate inference by both the baseline model and our DCL. Conversely, the two objects in Figure 2(b) share similar auditory and static information but differ significantly in dynamic information. Our method accurately infers the answers to both questions. This is especially notable in the case of the second question, which requires dynamic information for reasoning, thus highlighting the efficacy of our DCL in the decoupling of static and dynamic elements.

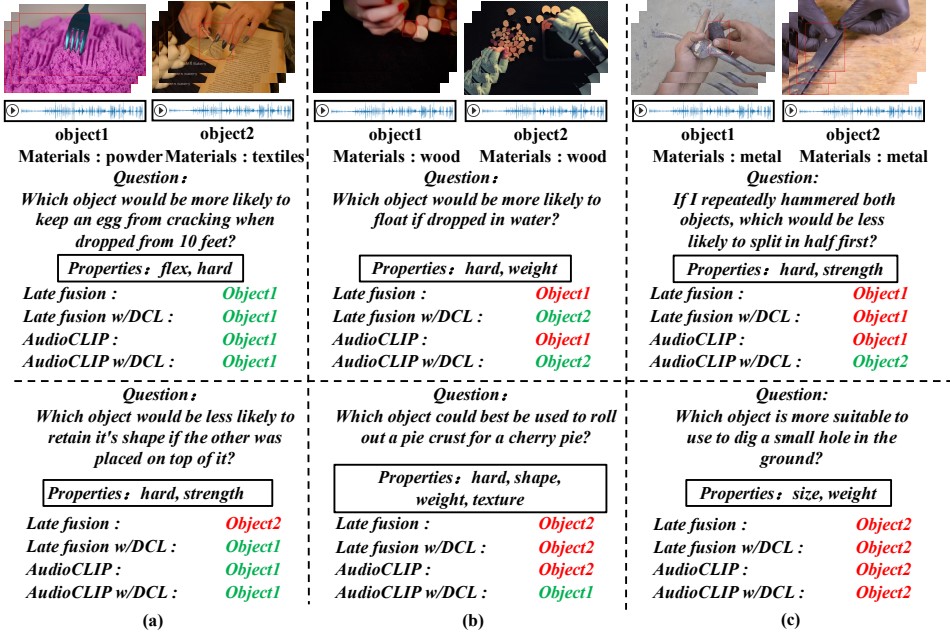

Figure 2: Qualitative Results of baseline w/ and w/o our DCL, where "Material" refers to the material of the object, and "Properties" refers to the physical properties that the question focuses on. The correct answers are depicted in green while the incorrect ones are depicted in red.

In Figure 2(c), only AudioCLIP enhanced with our DCL accurately answers the first question, while the other models fail. This discrepancy arises from inaccurate labeling within the dataset, where two objects appear in the same video, and the bounding box appears to have been assigned to the wrong object. By integrating a more robust object detection model and fine-grained annotations, our proposed DCL can circumvent such issues.

## 4.3 Ablation Study

In this section, we perform the ablation studies to examine the effectiveness of each of our proposed modules, including the Disentangled Sequential Encoder, physical knowledge relationship matrix, and counterfactual relation intervention in the Counterfactual Learning Module.

**Disentangled Sequential Encoder (DSE).** Table 1(b) shows the performance comparison of our proposed DSE with different baselines. And we can see AudioCLIP can achieve the best results. After incorporating DSE, all baselines are able to achieve significant improvements in PACS and PACS-Material. However, adding an MLP module with the same parameters as DSE did not yield such improvement, indicating that the proposed DSE can effectively enhance the ability of physical characteristic expression in video features.

**Counterfactual Learning Module (CLM).** Table 1(c) presents the ablative analysis of two designed parts in the Counterfactual Learning Module (CLM). After adding the physical knowledge relationship denoted as affinity matrix $A$, all baselines showed an improvement in performance. Similarly, the same result can be clearly found when adding the Counterfactual Relation Intervention denoted as 'C'. Additionally, the same conclusion can be drawn from the PACS-Material, demonstrating that both parts of our Counterfactual Learning Module can independently improve results and achieve optimal performance when used simultaneously.

## 4.4 The Analysis of Disentanglement Learning

In this section, we carry out additional experiments and analysis focusing on the decoupled static and dynamic factors obtained through the proposed Disentangled Sequential Encoder. Specifically, we utilize separated individual factors to predict the object's material information, excluding multi-modal

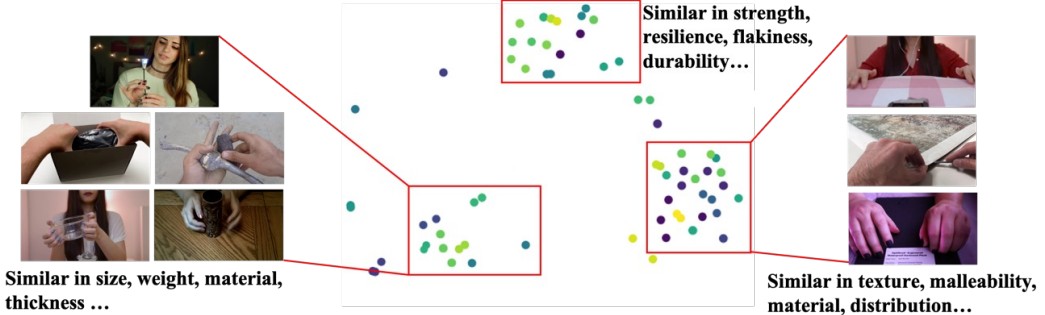

Figure 3: The t-SNE visualization of the obtained dynamic factors in the latent space of video samples in the test set. The points located close to each other as a cluster exhibit similar or identical actions, such as rubbing with the hand (left) and friction with the hand (right), while such two clusters are pulled away showing they have completely different dynamic information.

feature fusion. For the captured dynamic factors, we present the t-SNE visualization of various video samples to illustrate the distribution of these dynamic factors within the latent space.

**Analysis of static factor.** The material classification results are shown in Table 1(d), where "w/ static" and "w/ dynamic" respectively signify reasoning using only decoupled static factors and dynamic factors, while "w/ static, dynamic" denotes using both static and dynamic factors simultaneously. It can be observed that combining the static and dynamic factors significantly improves the accuracy of reasoning for material problems, compared to using only dynamic factors. This indicates the crucial importance of static factors in material property inference. When predicting using only static factors, there is no significant reduction in accuracy, confirming the adequacy of static factors in providing sufficient material-related information that is not captured by dynamic factors. This also implies that the static and dynamic factors contain little overlapping information and validate the effectiveness of our comparative estimation of mutual information (MI).

**Analysis of dynamic factor.** For dynamic factors, we decoupled the test set videos and showed the t-SNE visualization of the results, as shown in Figure 3. It can be observed that there is a certain clustering phenomenon in the latent representation of dynamic factors. By displaying the corresponding videos of the data points, we found that the points on the left panel of the figure mostly represent actions involving rubbing with objects by hand. In this case, the visual features are often insufficient to fully represent the characteristics of the object due to its thickness or weight, and the audio information is often the main basis for human reasoning. On the other hand, the points on the right side of the figure correspond to actions involving the friction of objects. Such actions can reveal more dynamic properties of the object, such as malleability. In this case, the dynamic information in the video is often the main basis for human reasoning. Through the above analysis, we can see that for different types of dynamic information, humans will also focus on different modalities of information. This is helpful for future improvements of our reasoning models.

## 5   Conclusion

In this paper, we introduced a method named Disentangled Counterfactual Learning (DCL) for physical audiovisual commonsense reasoning, in which a Disentangled Sequential Encoder was designed to decouple the video into static and dynamic factors represented as time-invariant and time-varied information related to actions, respectively. Additionally, we model the physical knowledge relationship among different target objects as an affinity matrix and apply counterfactual relation intervention to guide the model to emphasize the physical commonalities among different objects. Designed as a plug-and-play component, our method can be readily incorporated and has demonstrated its potential to significantly enhance multiple baselines.

## Acknowledgement

This work was partly supported by the Funds for Creative Research Groups of China under Grant 61921003, the National Natural Science Foundation of China under Grant 62202063, Beijing Natural Science Foundation (No.L223002), the Young Elite Scientists Sponsorship Program by China Association for Science and Technology (CAST) under Grant 2021QNRC001, the 111 Project under Grant B18008, the Open Project Program of State Key Laboratory of Virtual Reality Technology and Systems, Beihang University (No.VRLAB2022C01).

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
