# Disentangled Counterfactual Learning for Physical Audiovisual Commonsense Reasoning
## *Supplementary Material*

1 The supplementary material provides detailed implementation information on the baselines used for
2 comparison, as well as various analyses, such as hyper-parameters, model size and training time,
3 audio disentanglement learning, static factors, dynamic factors, and physical knowledge relationships.
4 Moreover, we show more visualization results in experiments.

## 1 Implementation Details of Compared Baselines

6 **LateFusion.** Following PACS [1], we used a pre-trained Debert-a-V3-Large [2] as the text encoder
7 to encode all questions into $\mathbb{R}^d$, where $d = 768$, which was saved during the training. We did not
8 apply any data augmentation to the text and extracted the text embeddings from the <CLS> token of
9 the text model's output layer (pre-pooler). The pre-trained model can be downloaded from here[1].

10 For videos, we downsampled them as input for the model. We used the ViT/B-16 model per-trained
11 on ImageNet-21k provided by HuggingFace[2] to extract features from the video frames. Following
12 the video augmentation steps used in the pre-trained model, we began with video frames of size 252
13 × 252 and randomly selected 8 evenly spaced frames. We then cropped the same 224 × 224 from
14 each frame and randomly flipped the images horizontally with a probability of 0.5.

15 For audio, we used the pre-trained AST (Audio Spectrogram Transformer) [3] model with a time and
16 frequency stride of 10 and weight averaging that was pre-trained on the full AudioSet [4]. The model
17 can be downloaded from here[3]. We followed the audio augmentation steps for pre-trained AST on
18 AudioSet [4] and PACS [1] using frequency and time masking [3]. We utilized 128 mel bins with a
19 target length of 1024. Then, we masked a band of size 48 in the frequency domain and a crew of size
20 144 in the time domain. Finally, we normalized the spectrogram as spec = (spec + 4.26)/(4.57*2) and
21 added random noise.

22 During training, we use a simple grid search and set the learning rate to $5 \cdot 10^{-4}$, the weight decay to
23 $5 \cdot 10^{-5}$ and the batch size to 64 (20GB of GPU memory was used). We trained the model for 40
24 epochs with early stopping, and we freeze all backbone layers and only use trainable MLPs layers to
25 fuse multimodal information.

26 **CLIP.** For CLIP [5], we only used the video frames $X^v = \{X_1^v, X_2^v, ..., X_T^v\}$ as input, where $T$
27 refers to the number of video frames and used the same ViT/B16 as the backbone network, along with
28 the same video preprocessing and augmentation, to obtain features $X = [\ X_s^v \ ^|_| \ X_z^v \ ]$ for objects.
29 To ensure a fair comparison, we used the fusion and optimization method as same as Latefusion.
30 Additionally, we use learning rate of $1 \cdot 10^{-4}$ with weight decay of $1 \cdot 10^{-5}$ after a simple grid search.

---

[1] https://huggingface.co/microsoft/deberta-v3-large

[2] https://huggingface.co/google/vit-base-patch16-224-in21k

[3] https://github.com/YuanGongND/ast

Submitted to 37th Conference on Neural Information Processing Systems (NeurIPS 2023). Do not distribute.

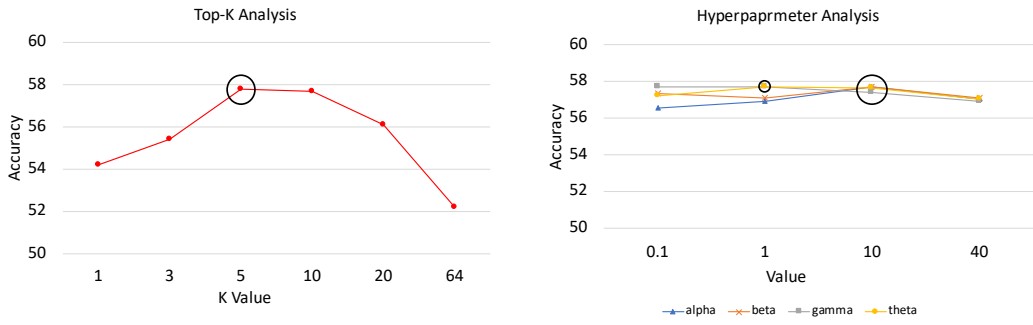

Figure 1: Hyper-parameters Analysis of our DCL.

**AudiCLIP.** For AudioCLIP [6], we used both audio $X^a$ and video $X^v$ as input. To ensure a fair comparison, we used the preprocessing and augmentation method as same as Latefusion. For hyperparameters, we used the same configuration as CLIP. We used ESRestNet(X)t-fbsp [7] as the backbone network for audio feature extraction. During training, we froze all layers of the backbone network, and only use trainable MLP layers to fuse multimodal information.

**UNITER.** In experiments, we utilized UNITER pre-trained on NLVR2 dataset [8] for feature extraction and fusion. The pre-processed sequential video information $X^v$ was used as input. In our experiments, UNITER's pair setup was applied to handle the input object-question pair as two independent text-video pairs. The concatenation of the output of both [CLS] in UNITER was regarded as $\hat{Y}_{X,A_X}$, as mentioned in the main paper. We use the learning rate of $1 \cdot 10^{-5}$ and a weight decay of 0.01.

## 2 Hyper-Parameters Analysis

Our proposed model consists of two main modules: Disentangled Sequence Encoder (DSE) and Counterfactual Learning Module (CLM). Specifically, in DSE, we used a hidden layer size of 256 for Bi-LSTM and set $\gamma = 1$, $\alpha, \beta = 10$, and $\theta = 1$. In CLM, $\tau = 2$ and $k = 5$ were used when calculating similarities and constructing the physical knowledge relationships. The hyper-parameters for both DSE and CLM were kept consistent when incorporated with each baseline. Figure 1 shows the hyper-parameter analysis of Late Fusion w/ DCL, where the left part illustrates the relationship between the Top-K value and the corresponding accuracy, with k taking values of 1, 3, 5, 10, 20, and 64. When k=1, it means that the object's physical properties are only related to itself, while k=64 represents the physical properties of the object that are related to all objects within the batch. From the figure, it can be clearly observed that when we set K as 5 the model can achieve the best performance, indicating that an appropriate K can help improve the ability to explore the common physical properties. Meanwhile, a value of k that is too large can introduce excessive noise while too small is insufficient to exploit the relevance among objects and both result in decreased accuracy. The right part in Figure 1 shows the results when different values of $\alpha$, $\beta$, $\gamma$, and $\theta$ are adopted, demonstrating that the model is insensitive to these hyperparameters, indicating its robustness. The black circular markers in the figure indicate the parameter values that were ultimately used.

## 3 Model Size and Training Time

Table 1 reports the model size and training time of our proposed method and baselines, by using an Intel 6226R CPU and an NVIDIA RTX 3090 GPU, in terms of the time required for a single epoch of training. It can be observed that the increase in training time is acceptable, and the additional memory usage is also within a controllable range.

## 4 Analysis of Audio Disentanglement Learning

In this section, we will analyze the effectiveness of our proposed Disentangled Sequential En-coder (DSE) on the input audio data. As described in Section 3.1 in our paper, we represent audio

| | Model Size | Train Time |
|---|---|---|
| Latefusion | 170.2M | 1,214s |
| Latefusion w/ DCL | 189.4M | 1,317s |
| AudioCLIP | 230.1M | 1,545s |
| AudioCLIP w/ DCL | 242.6M | 1,628s |

Table 1: The analysis of the model size and training time

| Baseline Model | Accuracy (%) | |
|---|---|---|
| | PACS | PACS-Material |
| Late Fusion [9] | $55.0 \pm 1.1$ | $67.4 \pm 1.5$ |
| Late Fusion w/ DSE-audio | $56.9 \pm 0.5$ | $68.1 \pm 0.4$ |
| AudioCLIP [6] | $60.0 \pm 0.9$ | $75.9 \pm 1.1$ |
| AudioCLIP w/ DSE-audio | $\mathbf{61.5 \pm 0.8}$ | $\mathbf{76.0 \pm 0.7}$ |

Table 2: Performance comparison between our proposed DSE-audio and existing baseline methods.

features as a sequence and extract them as sequence data represented by $X^a = \{X_1^a, X_2^a, ..., X_T^a\}$, where $T$ denotes the number of audio time steps. It should be noted that, in the audio decoupling experiment, we did not decouple the video but processed it through averaging.

**Quantitative Results.** As shown in Table 2, we compare our method with other baseline methods. Since only Latefusion [9] and AudioCLIP [5] take audio as input, we compared our method with both of them. It can be observed from Table 2 that using only DSE-audio, Late Fusion achieved an absolute improvement of 1.9%, while AudioCLIP achieved an absolute improvement of 1.5% on the PACS dataset. This indicates that our proposed DSE-audio method has a significant impact on audio decoupling. The same conclusion can also be drawn from the results on the PACS-Material dataset, which demonstrates the effectiveness and superiority of our DSE as a plug-and-play method that can achieve excellent performance on various datasets.

# 5    More Visualization Results

As shown in Figure 2 and Figure 3, we present more visualization results comparing our proposed method with other baseline models. It can be seen from the figures that our proposed DCL method outperforms the original method.

# 6    Analysis of Material and Question Properties

Figure 4 and Figure 5 show the results of object accuracy for per materials and per-property related to each question, respectively. From Figure 4, we can see that after using the proposed DCL, there was an improvement in object accuracy for all materials, especially for *Rubber*, *Plastic*, and *Metal* show the most prominent improvement. This is because our proposed DCL is capable of capturing clean dynamic features, which is essential in learning the physical properties of objects that often require dynamic movements. Based on Figure 5, after incorporating our proposed DCL, the accuracy of all properties related to questions improved, especially for Flexibility and Weight properties s the most significant improvement. The reason may attribute to the fact that these questions require more dynamic features for accurate judgment, and therefore, the usage of our proposed DCL led to a significant improvement in performance.

# 7    Analysis of Static Factors

Figure 7 and Figure 8 show the results of our proposed method using only static factors and other baselines. As shown in Figure 7, it can be observed that our model performs better than the baseline in terms of "Stone", "Glass", and "Textiles" while underperforming on "Metals", probably stems from that metals require more audio information for support. This finding demonstrates the advantage of the decoupled static factors in material classification. Figure 8 illustrates the performance of our

model on questions related to different physical properties, where it performs well w.r.t "texture", "shape", and "size", indicating the helpfulness of the decoupled static factors in these properties.

# 8 Analysis of Dynamic Factors

In Figure 6, we show a few additional examples of clustering using dynamic factors. It can be observed that the decoupled dynamic factors represent similar or related action information.

# 9 Analysis of Physical Knowledge Relationships.

Figure 9 presents the visualization of the physical knowledge relationships captured by our dynamic factors with the affinity matrix $A$, and the corresponding top-5 results are displayed. As shown in the figure, we can find that the selected top-5 video in the top row and the bottom row in the red box are actions that are similar or related to the given sample video, indicating the effective discovery of common physical relevance through our physical knowledge mining, as mentioned in our main paper.

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

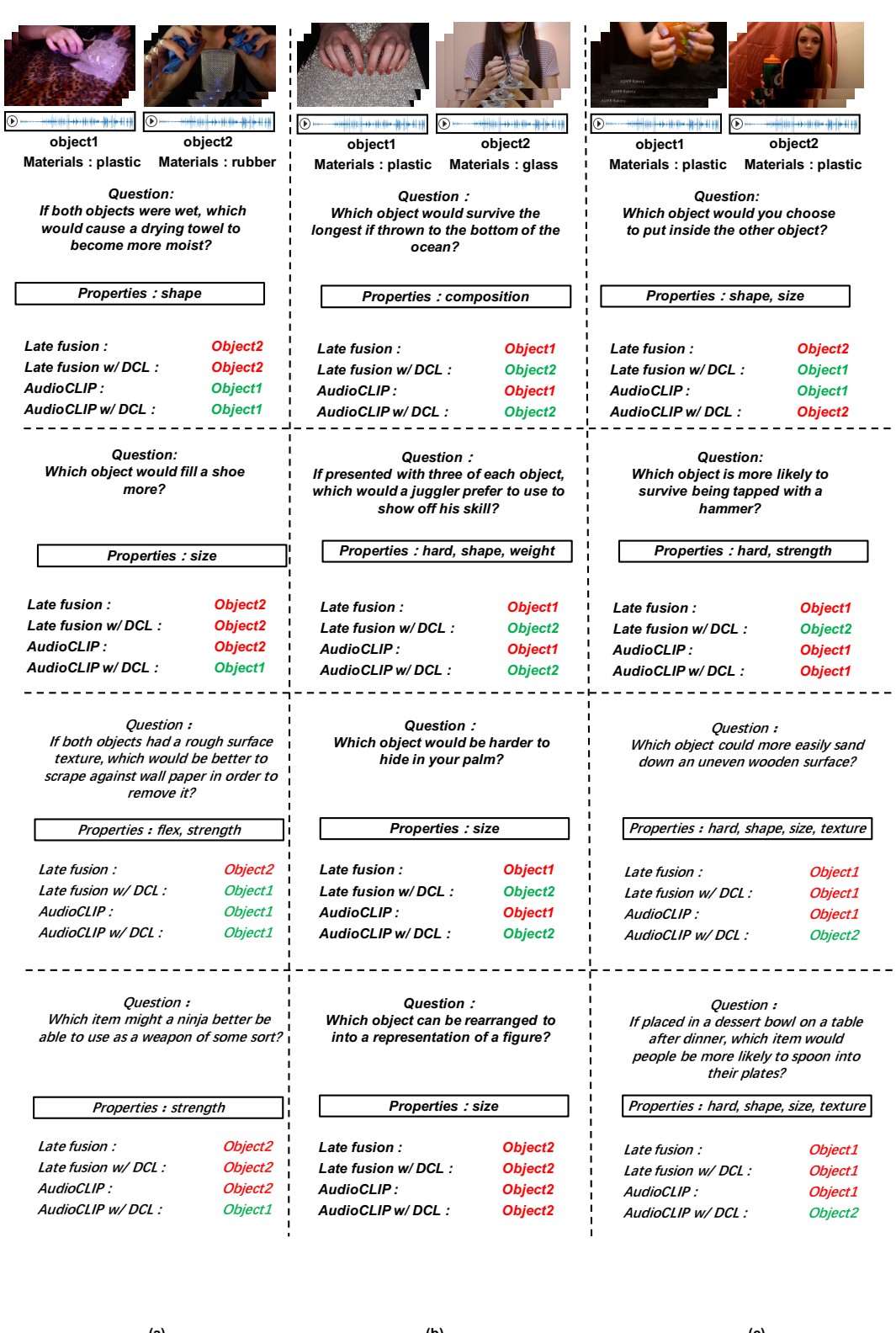

Figure 2: Comparison between our proposed method and existing baseline methods, where 'w/ DCL' indicates the baseline incorporated with our proposed DCL method.

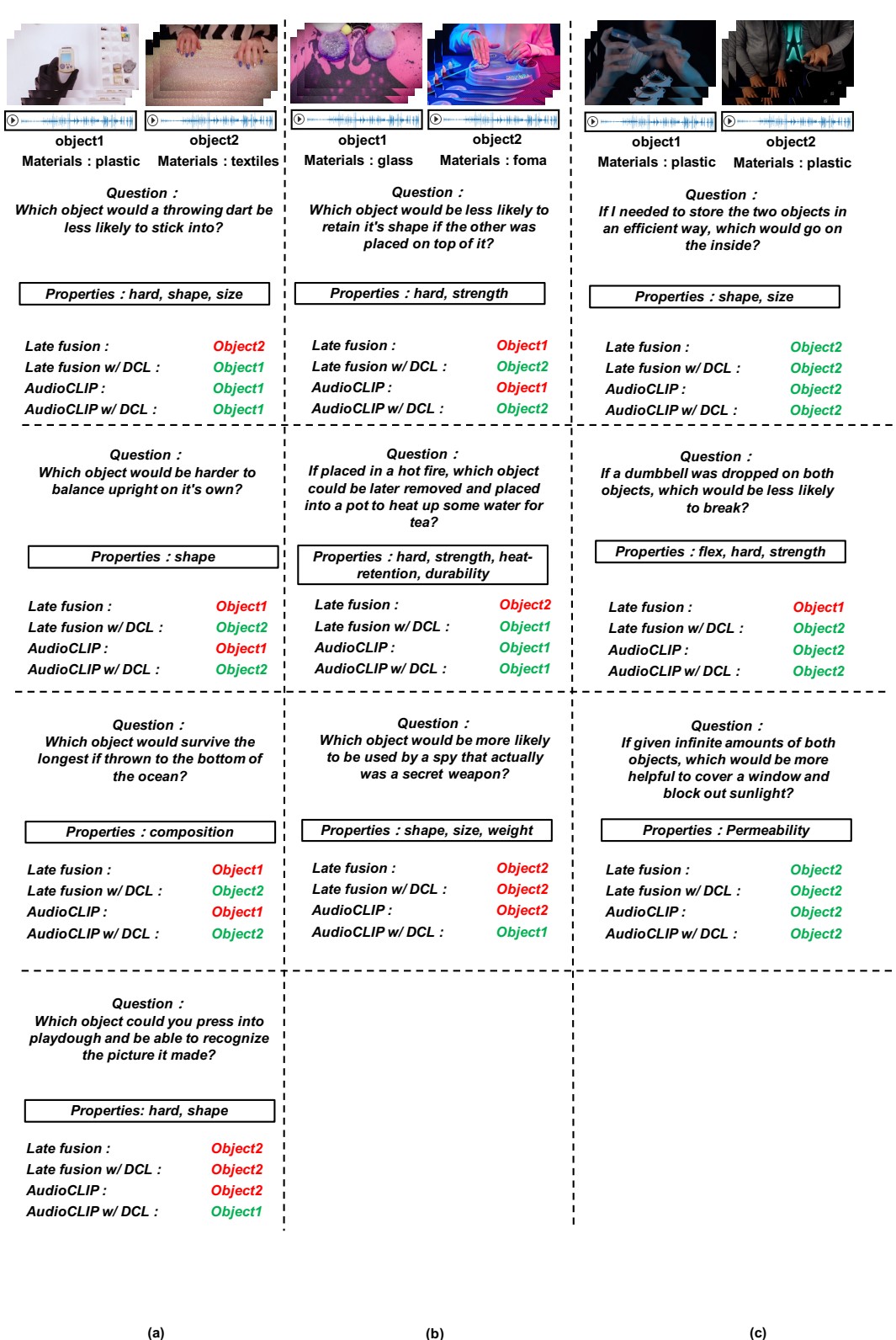

Figure 3: Comparison between our proposed method and existing baseline methods, where 'w/ DCL' indicates the baseline incorporated with our proposed DCL method.

Per-material accuracy analysis

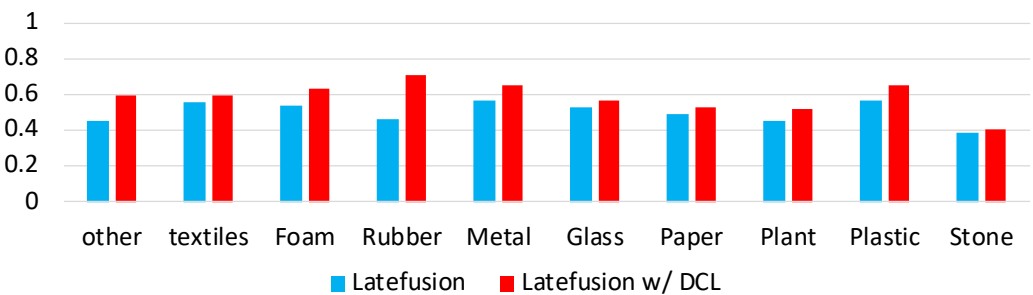

Figure 4: Accuracy results of per material object.

Per-property related to each question

Figure 5: Accuracy results of per properties related to each question.

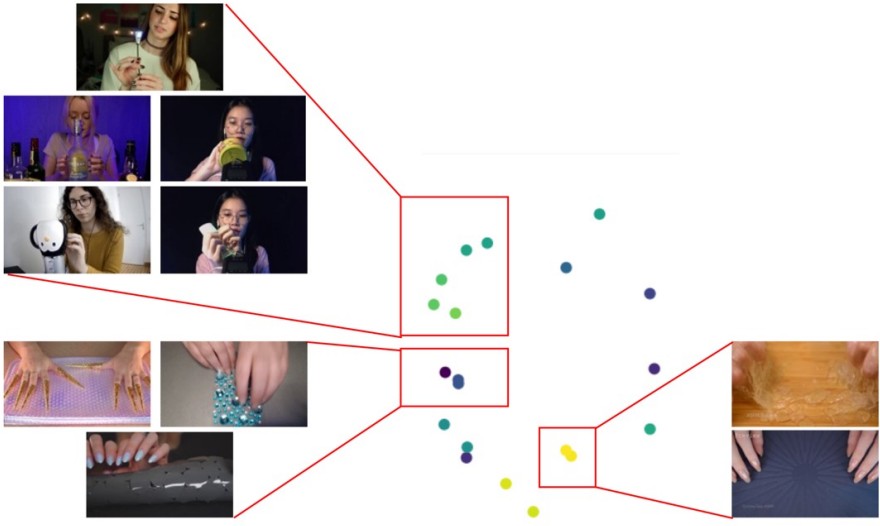

Figure 6: The t-SNE visualization of the obtained dynamic factors in the latent space of video samples in the test set.

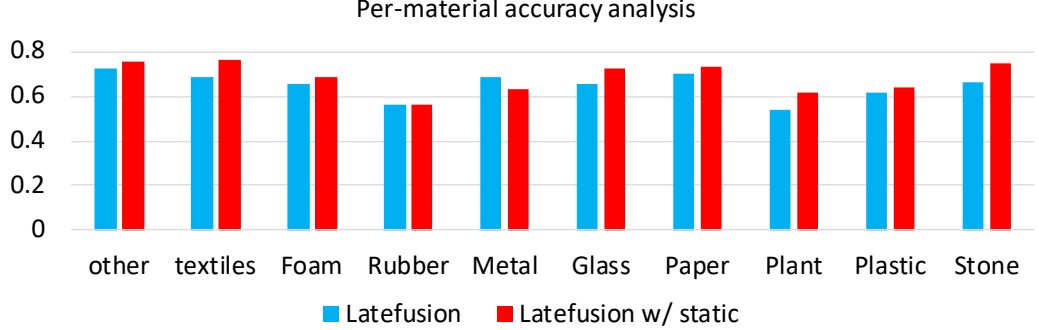

Figure 7: Accuracy results of per material object.

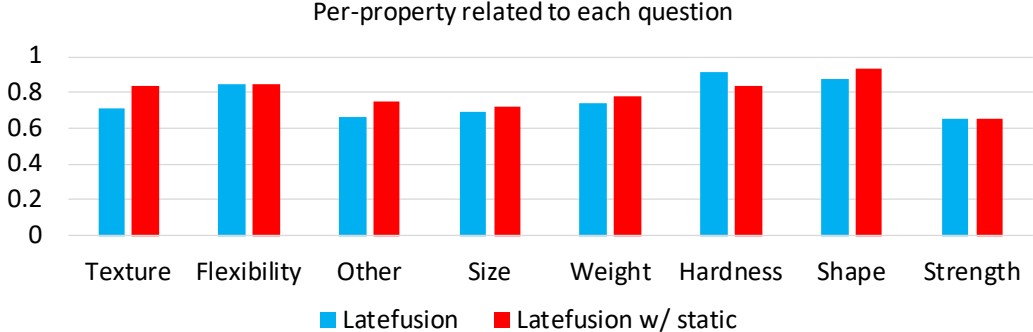

Figure 8: Accuracy results of per properties related to each question.

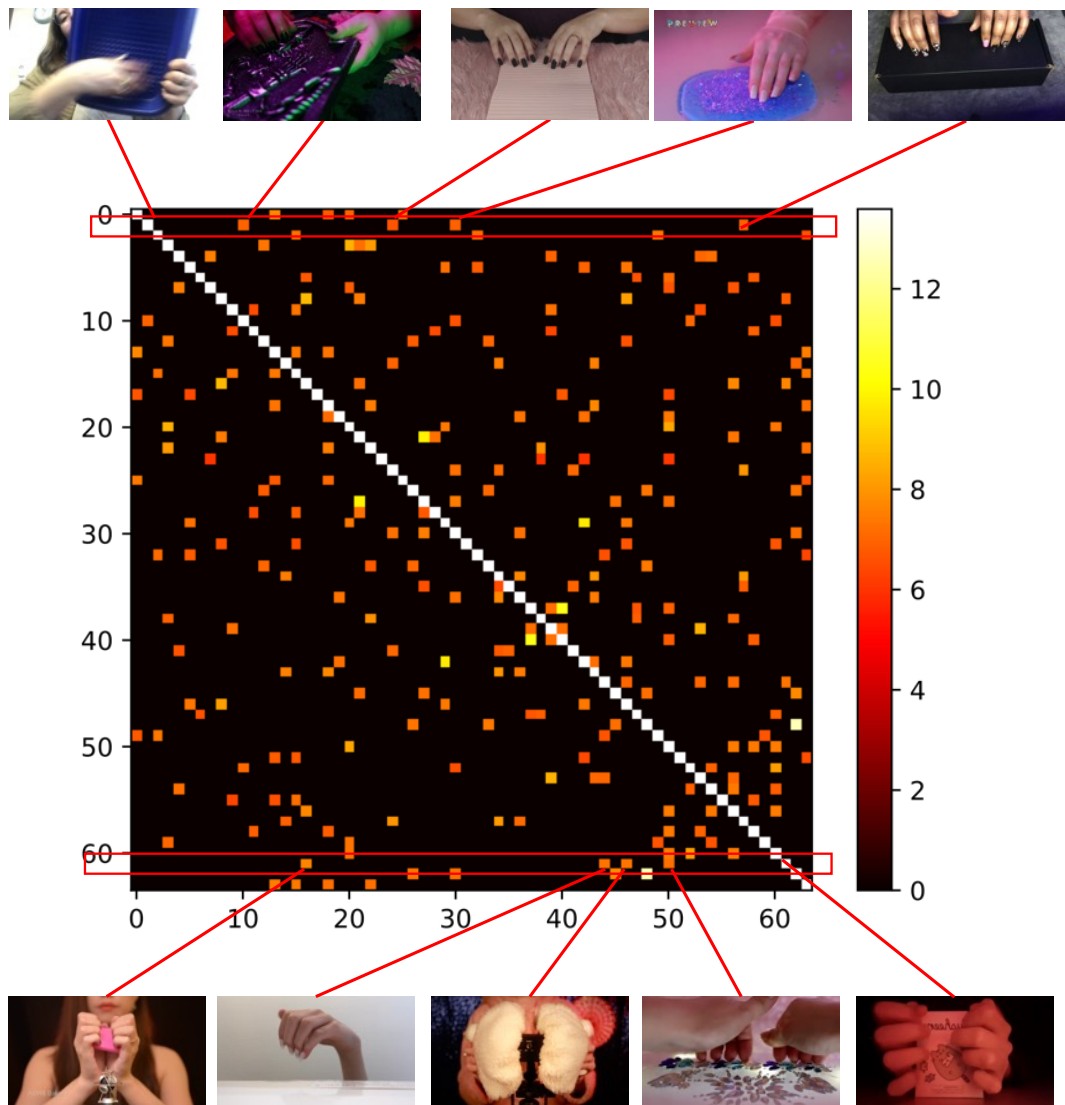

Figure 9: Visualization of Physical knowledge relationship.