# OpenReview forum: "Disentangled Counterfactual Learning for Physical Audiovisual Commonsense Reasoning"
_NeurIPS.cc/2023/Conference — NeurIPS 2023 poster_

### Official Review · Reviewer_PAiU · 2023-07-05

**Soundness:** 3 good
**Presentation:** 3 good
**Contribution:** 2 fair
**Rating:** 5
**Confidence:** 3

**Summary:**

This paper presents a groundbreaking Disentangled Counterfactual Learning (DCL) approach for physical audiovisual commonsense reasoning. The main objective of the proposed method is to infer objects' physics commonsense based on both video and audio inputs, effectively mimicking human reasoning abilities. To address the limitations of existing methods in utilizing the diverse characteristics of multimodal data and lacking causal reasoning abilities, the authors introduce the DCL method. Contributions: 1. introducing a novel DCL approach that leverages disentanglement and causal reasoning to improve multimodal data utilization and achieve remarkable performance. 2. propose a novel Counterfactual Learning Module to model physical knowledge relationships. 3. the proposed DCL method is designed as a plug-and-play module, making it adaptable for integration into various baseline models.

**Strengths:**

The paper introduces an innovative framework for tackling the challenging task of physical commonsense reasoning. The authors meticulously design a disentangled sequential encoder and a counterfactual learning module, both of which contribute to the success of the proposed model in addressing this unique visual question answering (VQA) problem. Importantly, the model's modular nature allows for seamless integration with various baseline approaches, thereby enhancing its versatility.

The experimental evaluation conducted in this study is relatively enough, demonstrating the effectiveness of the proposed method. The results showcase a significant improvement in accuracy on the PACS dataset, validating the model's ability to handle complex reasoning tasks. Moreover, the authors present qualitative results that vividly illustrate how their approach enhances material reasoning performance, further reinforcing the practical relevance and value of their work.

In terms of the paper's presentation, the English writing is commendable for its clarity and accessibility. The authors effectively convey their ideas, making it easy for readers to grasp the core concepts and understand the technical details without unnecessary complexity.

**Weaknesses:**

(1) The main experiment is limited to the PACS dataset and a specific physical audiovisual commonsense reasoning task. While the proposed method is presented as a plug-and-play module, it would be valuable to explore its generalizability to other VQA tasks and unseen data distributions. Considering the versatility of baseline models like CLIP, it would be worthwhile to investigate whether this module can be effectively applied in diverse scenarios.

(2) The ablation analysis is not fully convincing. As a plug-and-play module, it is crucial to clarify that the observed improvement is attributed to the unique design of the proposed method rather than an increase in the number of parameters. To strengthen the argument, additional quantitative experiments could be conducted, such as replacing the Disentangled Sequential Encoder (DSE) with a trivial naive module and comparing the results.

(3) The tables presenting the results of the quantitative experiments are not well displayed. To enhance clarity and readability, improvements could be made in the formatting and organization of the tables. Additionally, the t-SNE visualization in Figure 3 could benefit from displaying more distinct clusters and additional samples. The current arrangement of Figure 3 lacks persuasiveness, as the clusters on the 2D space are not significantly distinguishable from each other.

**Questions:**

 You can conduct more quanlitative experiments according to the weaknesses.

**Limitations:**

Limitations have been discussed in the paper

---

> ### Author Rebuttal · Authors · 2023-08-10
>
> # Response to Reviewer  PAiU
> Thank you for your time and valuable comments, below are our responses to your concerns:
> ## Q1: Applications in diverse scenarios.
> This is a question worthy of exploration. Our module is equally applicable to other scenarios. Below are our results on the Visual Commonsense Reasoning (VCR) dataset [1].
> | Model   | Q→A  | QA→R | Q→AR |
> |-----------|:------:|:------:|:------:|
> | LateFusion            | 40.2 | 35.6 | 10.1 |
> | LateFusion w/DCL | 45.7 | 40.5 | 20.1 |
> | CLIP                    | 46.5 | 37.4 | 15.3 |
> | CLIP w/ DCL           | 50.0 | 48.5 | 22.5 |
>
> According to the results, our method has led to improvements for both Latefusion and CLIP across multiple performance metrics showing the versatility of our proposed method.
> ## Q2: Replacing DCL with a trivial naive module.
> Thank you for your suggestion. We conducted the suggested experiments by replacing DCL with four layers of MLP . The results are as follows:
> | Model       | PACS |  PACS-Material |
> |--------------------|:----------:|:------------------:|
> | Audio CLIP         | 60.0±0.9 | 75.9±1.1         |
> | AudioCLIP w/ MLP   | 59.9±0.6 | 73.8±0.6         |
> | AudioCLIP w/DCL    | 63.2±0.8 | 76.2±1.4         |
>
> 'AudioCLIP w/ MLP' is the result of replacing DCL with MLPs on top of the AudioCLIP. The results clearly indicate that, even with comparable parameter settings, 'AudioCLIP w/ MLP' remains unable to match the performance of 'AudioCLIP w/DCL', affirming the effectiveness of our designed DCL module.
> ## Q3: Improvements of Table.1 and a new Fig.3.
> Thank you for your suggestion. We will  improve the tables in our paper to improve clarity and readability. To address your concerns about Fig. 3, we will replace it with the new t-SNE plot from the rebuttal materials Figure 1.
>
> [1] From Recognition to Cognition: Visual Commonsense Reasoning CVPR 2018

---

> ### Author Response · Authors · 2023-08-19
>
> Thanks again for your insightful suggestions and comments. Please let us know if our responses have addressed the issues raised in your reviews. We hope that our clarifications and additional results address your concerns and convince you of the merits of our work. We are happy to provide any additional clarifications or experiments that you may need.
>
> Thank you for your time again!
>
> Best, Authors

---

### Official Review · Reviewer_FCQF · 2023-07-06

**Soundness:** 3 good
**Presentation:** 4 excellent
**Contribution:** 3 good
**Rating:** 8
**Confidence:** 3

**Summary:**

The work proposes an approach to separate object and action information from videos to improve the model's reasoning capabilities. The proxy-task of audio-visual question answering is used to train the model commonsense concepts of the physical world. Their contribution focuses on learning from introduced counterfactual examples.
They aim to maximise mutual information between input data and static and dynamic scene factors while minimizing the mutual information between those two factors.

**Strengths:**

The paper presents beyond a good intuition also a detailed mathematical definition of all concepts. Both can be followed well. A fair amount of relevant and current baseline methods were used for comparison. The approach seems to be lightweight enough to be trained on one single GPU within reasonable time.

The ablation studies seem adequate and useful to understand the success of the disentanglement and the performance boost over baselines.

**Weaknesses:**

Compared with datasets in other fields the used dataset seems just about big enough to test the approach however a test set in the PACS-Material dataset of 152 objects could potentially lack robustness. Other selected 152 objects may change the result by a large margin. However, in this specific domain it is probably hard to generate larger datasets and comparable work uses the same datasets. K-fold cross evaluation could help here to evaluate the result better.

There is a clear trend that the method performs well in Table 1, however given the error the results could be more impressive. Again, K-fold cross validation with different splits could help.

The paper mentions a few times that it can be used as a plugin to improve performance on multimodal fusion tasks, however due to the Q&A being an integral part I don't see how this can generalize to e.g. benefit tracking. Due to their claim it would be nice if the authors could illustrate examples of potential tasks that could be improved with this method.

**Questions:**

In Figure 1. What is the referenced object? There are screws and a glass. What if there are several objects in the video but the question is ambiguous because it refers to only one object?

Extracting audio as non-sequence seems odd. How long is the audio instance? What about objects that change their sound characteristically? E.g. think about a laundry machine and a tumble dryer. Both may make some sounds which are over a longer time similar but then the laundry machine starts spinning and you can tell them apart. With such sounds it's also hard choose the part of the sound to encode. How is the length choosen? Does it just cover exactly all the video frames?

How does the approach deal with different lengths of videos? Is this just not regarded?

It is hard to find an information of how long the video sequence or audio sequence is in general. In the supplementary material, details are given for the implementation of LateFusion and there it can be found that 8 evenly spaced frames are used but does that mean that every single frame is used or are there skipped frames? And what is the framerate?

In line 175 it should say standard and not stand.

The object in Figure 2, b) left side is hardly recognizable. It says the material is wood but is this a woodchip or a fake wood coin? It's very hard to see. And what kind of sound does this make? Are those woodchips dropped on a hard surface? It would be good to either choose another object from another image which is more illustrative or to write down somewhere what the reader actually should see.

In the "Analysis of dynamic factor" it is said two times, audio information is often the main basis for human reasoning or dynamic information is often the main basis for human reasoning. I would just leave out those broad claims which can not be answered by training a network. It reads to me like a very general remark without any citation or investigation to back it up and should probably left to the field of HCI.

In Table 1, how was the error calculated? Were different training runs performed with different random seeds?

**Limitations:**

The small dataset test set could probably be mentioned or defended somewhere for people who are not right inside the domain. The broader impact section feels like being included just as exercise but could be removed imo. The argument "if our algorithm is implemented on a robot and malfunctions this could be bad" is too general to be useful.

---

> ### Author Rebuttal · Authors · 2023-08-10
>
> # Response to Reviewer  FCQF
> Thank you for your time and valuable comments, below is our response to your review:
> ## Q1: K-fold cross validation.
> Thank you for your suggestion. We conducted the experiments on the PACS-Material subset using the K-fold cross-validation approach you mentioned. The results are as follows:
> | Model                | PACS Material |
> |----------------------|:--------------:|
> | Late Fusion w/ DCL   | 66.2±0.6     |
> | CLIP w/ DCL          | 74.9±0.8     |
> | AudioCLIP w/ DCL     | 75.4±1.0     |
> | UNITER w/ DCL        | 75.2±0.7     |
> ## Q2: Potential tasks that could be improved with our method.
> Our proposed method can be extended to other multi-modal scenarios, such as decision-making in the context of autonomous driving. By incorporating our proposed module, the reasoning capability of the decision model can be enhanced. Similarly, in the behavioral reasoning process of intelligent robots, our approach can highlight the role of certain features in the decision-making process, thereby improving interpretability.
> ## Q3: Questions about our paper.
> Regarding your questions, we will address them individually:
>
> 1. In general, we consider objects that interact with human hands in the video and produce sound to be the referenced objects. In Fig.1, since the glass is held by a person and produces sound when struck, we consider glass to be the referenced object.
>
> 2. In the PACS dataset, the average audio length is 7.6 seconds. In PACS, both the objects in the video and the audio are relatively singular, we encoded all the audio corresponding to the videos in our experiments. Considering the temporal pattern of audio data, for cases involving changes in audio, we can treat audio as sequential data for processing in the future. Great suggestions!
>
> 3. We uniformly downsampled videos of varying lengths to 252 frames using corresponding proportions.
>
> 4. We used the video format of 30 FPS. Based the downsampled 252 frames, we uniformly sampled 8 frames while skipping others. Later, we utilized ViT to extract features from each frame, and then utilized the averaged features as the video representation.
>
> 5. We appreciate your pointing out our errors. We will change "stand" to "standard" on line 175.
>
> 6. Thank you for pointing out the issue. We appreciate your suggestion and have adopted it by replacing Fig2 (b) with a new example. The new example is provided in rebuttal material Figure 4.
>
> 7. Indeed, as you mentioned, we have identified two relevant works in the field of HCI to support the importance  of the dynamic information [1,2].
>
> 8. As you mentioned, the errors in Table 1 are calculated by training with different random seeds and averaging the results to compute the error.
> ## Q4: Limitation about our method.
> We appreciate your valuable suggestion, and we will make modifications to the expression of this part in the revised version.
>
> [1] Enabling Voice-Accompanying Hand-to-Face Gesture Recognition with Cross-Device Sensing. CHI2023
>
> [2] FaceSight: Enabling Hand-to-Face Gesture Interaction on AR Glasses with a Downward-Facing Camera Vision. CHI2021

---

> > ### Comment · Reviewer_FCQF · 2023-08-14
> >
> > I read the response and weaknesses other reviewers pointed out. I think the authors responded very well to all points. Including audio is still a bit beside the mainstream and hard to make work. Even if the method does not improve by a large margin over the state of the art the evaluation with additional parameters seems to show there is an actual effect here. I think this work deserves to be accepted and discussed in the larger research community because the method is novel, interesting and potentially foundational to other results which may yield more impressive results.

---

> > > ### Author Response · Authors · 2023-08-19
> > > **Response to Reviewer FCQF**
> > >
> > > We are grateful to you for not only acknowledging our answer, but also for finding our updates satisfactory. We strongly believe that those suggested changes have made our submission stronger.

---

### Official Review · Reviewer_j4kk · 2023-07-07

**Soundness:** 2 fair
**Presentation:** 1 poor
**Contribution:** 2 fair
**Rating:** 5
**Confidence:** 3

**Summary:**

This paper proposes a novel Disentangled Counterfactual Learning (DCL) method for physical audiovisual commonsense reasoning. DCL consists of two main modules: Disentangled Sequential Encoder and Counterfactual Learning Module(CLM). Disentangled Sequential Encoder decouples videos into static (time-invariant) and dynamic (time-varying) factors in the latent space. The causal learning module augments the model’s reasoning ability by modeling physical knowledge relationships among different objects under counterfactual intervention. The experiments show that DCL can be flexibly integrated into baseline methods and improve their performance.

**Strengths:**

+ The designed DCL is a plug-and-play module that can be incorporated into baseline methods.
+ The proposed method reaches SOTA performance. The authors also conduct ablation studies to illustrate the effectiveness of each component in their method.

**Weaknesses:**

- This paper is poor-written. Some expressions are not consistent, e.g., "causal learning module" in the abstract and "counterfactual learning module" in the introduction.
- Why causal learning can help commonsense reasoning? Which concrete problem in audiovisual commonsense reasoning can causal learning address? Although the authors introduce causal learning to augment the model's reasoning ability, no deep analyses are provided in this paper. The authors can provide an example or toy experiments to illustrate their motivation.
- Figure 3 only shows the t-SNE visualization of dynamic factors. Can the authors show t-SNE of static factors and the comparison with baselines?
-  Why do the authors only use disentangled sequential encoder (DSE) in videos? Can audios be processed by a similar operation?

**Questions:**

Please see weaknesses.

**Limitations:**

The authors present the broader impact in this paper.

---

> ### Author Rebuttal · Authors · 2023-08-10
>
> # Response to Reviewer  j4kk
> Thank you for your time and valuable comments, below is our response to your review:
> ## Q1: Some typos.
> We will revise the "causal learning module" on line 11 of the paper to "counterfactual learning module", and make sure that the entire paper is consistent.
> ## Q2: Why do we use causal learning?
> The motivation for using causal learning was primarily inspired by the following three aspects:
>
> 1. The binary labels in the dataset are insufficient to represent the real Ground Truth.
>
> Within the PACS dataset and audiovisual commonsense reasoning, binary labels do not truly correspond to the Ground Truth. For example, different interaction actions also result in distinct audiovisual features, labels are influenced by both the physical attributes (Ground Truth) exhibited by object's audiovisual features and the human-object interaction actions in the video.  In Fig3 (c) of the main paper, object1, when subjected to the same interaction action as object2, produces a similar sound (the sound of metal being rubbed). Addressing these challenges necessitates the utilization of causal learning to enhance the model's reasoning capabilities.
>
> 2. Learning from the correlation P(Y|X) can't  identify the causal effect from X to Y
>
> As mentioned in [1], if we only train the model based on the correlation P(Y|X) without understanding the confounding effect, regardless of the size of the training data, the model will never be able to discern the true causal effect from X to Y. We employ causal learning to guide the model in learning the true causal effect from audiovisual information to physical commonsense.
>
> 3. Using intervention to highlight the role of physical knowledge in reasoning
>
> Drawing inspiration from [2], the intervention in confounding factors can shed light on the significance of particular features within the reasoning. We utilize the physical knowledge similarity matrixs of decoupled video and audio features as confounders, through interventions, highlight the role of physical knowledge in reasoning, and makes the physical knowledge similarity matrix more reliable.
> ## Q3: Show t-SNE of static factors and the comparison with baseline.
> We appreciate your valuable suggestion. In our rebuttal material Figure 2 and 3, we have provided a comparison between the t-SNE plots depicting static factors and the t-SNE plots depicting video features of  the baseline (due to the fact that the baseline was not disentangled and only contains video features) . Figure 2 illustrates the t-SNE visualization of video static factors decoupled by AudioCLIP w/ DCL, while Figure 3 showcases the t-SNE visualization of video features by Audio CLIP.
> ## Q4: Disentangled sequential encoder (DSE) in audios.
> We indeed presented the experimental results of audio decoupling in our Supplementary. Please kindly see Supplementary Table 2 and the response to Reviewer QHKe's Q4 and Q5 for more details.
>
> [1] Causality: models, reasoning and inference. Judea Pearl 2000
>
> [2] Counterfactual Intervention Feature Transfer for Visible-Infrared Person Re-identification. ECCV2022

---

> ### Author Response · Authors · 2023-08-19
>
> Thanks again for your insightful suggestions and comments. Please let us know if our responses have addressed the issues raised in your reviews. We hope that our clarifications and additional results address your concerns and convince you of the merits of our work. We are happy to provide any additional clarifications or experiments that you may need.
>
> Thank you for your time again!
>
> Best, Authors

---

> > ### Comment · Reviewer_j4kk · 2023-08-20
> >
> > Thanks for the clarifications. I raise my score to borderline accept after reading authors’ response and other review comments.

---

### Official Review · Reviewer_QHKe · 2023-07-12

**Soundness:** 3 good
**Presentation:** 3 good
**Contribution:** 2 fair
**Rating:** 5
**Confidence:** 3

**Summary:**

The paper introduces a novel approach for physical audiovisual commonsense reasoning by Disentangled Counterfactual Learning (DCL). The authors propose a disentangled sequential VAE to separate static and dynamic factors in the visual latent space with an additional contrastive loss term. In addition, a causal learning module that leverages counterfactual intervention between different objects is used to enhance the learning of physical knowledge relations. The proposed modules could be easily plugged into existing baselines. The experiments on PACS dataset demonstrate that the proposed method could improve baseline methods.

**Strengths:**

1. The idea of modeling implicit physics knowledge relationships between various objects from audio-visual data is interesting and well-motivated.

2. The proposed method is a general module that could be plugged into any baseline. I could also see these modules are definitely not limited to being applied for audio-visual commonsense reasoning.

3. The paper is well-written and straightforward to understand.

4. Extensive experiments and analyses have been done to demonstrate the contribution of each proposed component.

**Weaknesses:**

1. While I like the general idea of the proposed modules, the final performance shows a minor increment (1.4-3.2%) compared to the baselines. With all these sophisticated designs of additional components, I would expect a larger gap in terms of performance, even though I believe audiovisual physical commonsense reasoning is a challenging task. If I understand correctly, the questions in PACS are binary (e.g., object 1 or 2). As I look into the results, the accuracy is about 60% which is not much better than a simple random guess. In the Supplementary material, there is a model size comparison between models with/without DCL. With the DCL, about 12M more parameters are introduced. Is it possible that the improvement is simply from these additional parameters? A simple verification would be adding the same amount of parameters via MLPs on top of the existing baselines and reporting the outcomes without using DCL.

2. In Section 3.2, Line 137, the authors mention the full proof could be found in Appendix, but it is missing.

3. The T-SNE plots in Figure 3 and the Supplementary material could be more informative. The description of each color in the plot is missing.

4. See Questions.

**Questions:**

1. In the current formulation, only the video features are represented as sequence data, but the audio feature is not. What is the reason for doing that? Based on the results, AudioCLIP actually works better than CLIP. This means probably audio plays a more important role in the reasoning. I guess having a disentangled sequential encoder for audio features and using contrastive loss could be possible.

2. Following the first question, the disentangled sequential encoder is for unimodal only (visual). I wonder whether adding a cross-modal mutual information term would be helpful if both audio and visual features are considered sequential.

3. As I mentioned in the weakness, the final performance is still around 60% accuracy, even with the proposed method. While I see some examples in qualitative results and analyses, like inaccurate labeling, I would like to ask what the main failure cases would be and the challenges to overcome.

4. Currently, the method is end-to-end training. I wonder whether it would be helpful if the disentangled sequence VAE is pre-trained first so that the static and dynamic factors are well-learned before applying counterfactual intervention.

---

> ### Author Rebuttal · Authors · 2023-08-10
>
> # Response to Reviewer QHKe
> Thank you for your time and valuable comments, below is our response to your comments:
> ## Q1: The impact of  more parameters.
> Thank you for your appreciation of our model. Audiovisual physical commonsense reasoning is indeed a formidable challenge (we will elaborate on the specific difficulties shortly).
>
> We added four layers of MLP with an equivalent number of parameters (12M) to replace our DCL into AudioCLIP, yielding the following results:
> | Model| PACS |
> |--|:--:|
> | Audio CLIP| 60.0 ± 0.9|
> | AudioCLIP w/ MLP| 59.9 ± 0.6|
> | AudioCLIP w/ DCL| 63.2 ± 0.8|
>
> The results clearly indicate that, even with comparable parameter settings, AudioCLIP w/ MLP remains unable to match the performance of AudioCLIP w/ DCL, affirming the effectiveness of our designed DCL module.
>
> ## Q2: Missing proof.
> The proofs presented in Section 3.2 are standard for sequential VAEs [1,2], so we omitted them for brevity. Please kindly see Appendix A in [1] for the full proofs. We will incorporate these proofs into the appendix of our final version.
> ## Q3: More information of the t-SNE plots in Figure.3 and the Supplementary.
> Thanks for the suggestion! We have included more comprehensive t-SNE visualizations in our rebuttal material (see Figure 1).
> ## Q4: The experiment of disentangled sequential encoder for audio features and using contrastive loss.
> Great suggestion! Following the baselines in PACS [3], we have abstained from treating audio as sequential data for fair comparison.
>
> However, we indeed apply a disentangled sequential encoder to audio features and results are shown in Table 2 of our Supplementary. Decoupling the audio led to a relative improvement of 2.5% in the model's performance. We will incorporating the results into the main paper.
>
> During training of the disentangled sequential encoder, we utilized contrastive loss, as outlined in Eq.(3)(4)(5) in the main paper, and the results also demonstrate its effectiveness.
> ## Q5: An experiment of adding a cross-modal mutual information term.
> Thanks for suggestions. Per your request, we simultaneously treat both visual and audio data as sequential data and disentangle them concurrently. Subsequently, we introduce cross-modal mutual information between the two modalities. The results are as follows:
> | Model|PACS|PACS-Material|
> |--|:--:|:--:|
> | AudioCLIP| 60.0±0.9|75.9±1.1|
> | Audio CLIP w/ DCL (visual)| 61.5±0.8 |76.0±1.4|
> | Audio CLIP w/ DCL (audio)| 63.2±0.8 |76.2±0.7|
> | Audio CLIP w/ DCL (visual and audio)|66.7±0.6 |79.4±1.1|
> | Audio CLIP w/ DCL (visual and audio) w/ MI| 65.5±0.8|78.6±0.5|
>
> As shown in the table, it is clear that simultaneous decoupling of both audio and visual modalities leads to superior results. However, we did not observe any additional enhancement when we included the mutual information term. We speculate that this is because visual and audio modalities could contain irrelevant information, and adding cross-modal mutual information could actually be detrimental to the results.
> ## Q6: What the main failure cases would be and the challenges to overcome.
> The challenges of our task can be summarized into three aspects:
>
> 1. Visual data contains a significant amount of noises.
>
> Within videos, there exists an abundance of noises such as diverse video backgrounds, and other irrelevant objects.  Extracting generic features using a simple visual encoder yields unsatisfactory results.
>
> Main failure case: In the main paper's Fig.1, in object1's video, besides the tall glass, the screws below serve as irrelevant objects, and the complex background of object2's video introduces noises.
>
> 2. The cause of audio generation is difficult to reason.
>
> Within audio, it is challenging to determine the relationship between actions and sounds. The same object, when interacted with different actions, can produce different sounds, thereby affecting the reasoning of its attributes. Extracting generic features using a single-modal audio encoder yields unsatisfactory results.
>
> 3. The questions' setting is challenging.
>
> For example, in the VQA2.0 [4] and VCR [5] dataset, questions such as "What color are her eyes?" necessitate a single round of reasoning to yield an answer. In contrast, within the PACS dataset and audiovisual commonsense reasoning, questions like "Which object would be less likely to retain its shape if the other was placed on top of it?" mandate the simultaneous comprehension of the physical attributes of two objects. This intricate configuration of multi-dimensional, multi-step reasoning compounds the difficulty. The binary labels in the dataset is insufficient to represent the real Ground Truth.
>
> Main failure case: In main paper Fig3 (c), the binary labels erroneously guide the model to believe that "objects that sound rough are not suitable for digging holes", whereas the actual ground truth is that larger objects are not suitable for digging small holes.
> ## Q7: Pre-trained VAE training for our method
> Thanks! We conducted additional experiments following your suggestion, and the results are as follows:
> | Model | PACS | PACS-Material |
> |--|:--:|:---:|
> | CLIP w/ DCL -End to End| 56.3±0.7 | 72.4±1.1|
> | CLIP w/ DCL -Pre-trained VAE | 55.2±0.3 | 70.1±1.5|
>
> where 'CLIP w/ DCL -Pre-trained VAE' is the method of first training VAE and then train other models, while the other one is our original end-to-end method. As shown in the table, the pre-trained VAE's performance is inferior to that of the end-to-end approach.
>
> [1]S3vae: Self-supervised sequential vae for representation disentanglement and data generation. CVPR2020
>
> [2]Contrastively disentangled sequential variational autoencoder. NIPS2021
>
> [3] PACS: A Dataset for Physical Audiovisual CommonSense Reasoning. ECCV2022
>
> [4] Making the V in VQA Matter: Elevating the Role of Image Understanding in Visual Question Answering. CVPR2017
>
> [5] From Recognition to Cognition: Visual Commonsense Reasoning. CVPR2018

---

> > ### Comment · Reviewer_QHKe · 2023-08-14
> >
> > I appreciate that the authors have conducted new experiments, and these results have addressed most of my concerns. Although the performance improvement is incremental, the proposed approach is generally novel and could benefit the community. With this consideration, I would like to raise my score to 'borderline accept'.

---

> > > ### Author Response · Authors · 2023-08-19
> > > **Response to Reviewer QHKe**
> > >
> > > We are grateful to you for not only acknowledging our answer, but also for finding our updates satisfactory. We strongly believe that those suggested changes have made our submission stronger.

---

### Official Review · Reviewer_ZZdS · 2023-07-13

**Soundness:** 3 good
**Presentation:** 2 fair
**Contribution:** 2 fair
**Rating:** 4
**Confidence:** 4

**Summary:**

This paper proposes a disentangled counterfactual learning (DCL) approach to solve physical audio-visual commonsense reasoning. This approach first decouples videos into static and dynamic latent features, and then uses a causal learning module to augment the model's reasoning ability. Authors show that this module can be used to augment any existing baselines.

**Strengths:**

The proposed approach is novel and inspiring for similar reasoning tasks. The approach section has detailed each sub-module of the approach. I also like the fact that this approach can be plugged into any existing work to improve its performance.

The qualitative examples in Fig. 2 show how baseline models augmented with the proposed approach performs better than those without.

**Weaknesses:**

The performance improvement against baselines is pretty marginal, especially it did not run Merlot Reserve with the additional module. I understand that it was because the computational resource is restricted. But still, since Merlot Reserve has a 10% improvement over the second-best baseline, it remains a question whether this proposed module still brings benefits to Merlot Reserve.

Some other comments:

1. Random performance (guessing by chance) should be shown in Table 1.
2. The reported baselines are baselines in the benchmark but that does not represent the current SOTA in this domain anymore. There have been many other works coming out after CLIP for example.
3. Late Fusion w/ dynamic [42] should be written as Late Fusion [42] w/ dynamic. And the same applies to other rows in Table 1.
4. You need a more concrete example to motivate the physical knowledge relationship module. It is unclear in the paper what kind of correlation between different samples are shared.

**Questions:**

My main question about the paper is regarding the performance margin and I'd appreciate authors' clarification on that.

**Limitations:**

The limitation of this work has not been discussed but the societal impact has been discussed in the last paragraph.

---

> ### Author Rebuttal · Authors · 2023-08-10
>
> # Response to Reviewer ZZdS
> Thank you for your time and valuable comments, below is our response to your review:
> ## Q1: Marginal Performance of our method.
> Merlot Reserve utilizes a large-scale private dataset (YT-Temporal-1B) and a specialized device (v3-1024 TPU) for training. Due to this limitation, we were unable to reimplement the experimental setup of Merlot Reserve.
> To address concerns about the performance gap, we evaluated the model ERNIE-ViL [1] as a replacement for Merlot Reserve. This is because ERNIE-ViL and Merlot Reserve (base) achieved comparable performance on the VCR task [3]. Additionally, we incorporated the SOTA model BLIP [2] after CLIP in our experiments. The results are as follows:
> | Model| PACS | Relative Improvement (%) |
> |----------------------|:-------------:|:-------------------------:|
> | Random| 50.4|-|
> | Late Fusion|55.0 ± 1.1|-|
> | Late Fusion w/DCL|57.7 ± 0.9| 4.9|
> | CLIP| 56.3 ± 0.7|-|
> | CLIP w/ DCL| 58.4 ± 0.8| 3.7|
> | AudioCLIP  | 60.0 ± 0.9  | -    |
> | AudioCLIP w/ DCL  | 63.2 ± 0.8  | 5.3  |
> | BLIP [2]   | 59.1 ± 0.5  | -    |
> | BLIP w/ DCL   | 61.5 ± 0.6  | 4.0  |
> | UNITER (Large)   | 60.6 ± 2.2  | -   |
> | UNITER (Large) w/DCL | 62.0 ± 2.4  | 2.3   |
> | ERNIE-ViL [1]    | 66.7 ± 1.1  | -    |
> | ERNIE-ViL w/ DCL | 70.4 ± 0.8  | 5.5  |
>
> As shown in the table, ERNIE-ViL outperforms the second-best baseline by 6%, and with our DCL module, it achieves an improvement of 5.5%. Although it cannot directly prove the effectiveness of our module on Merlot Reserve, based on the results of the six tested models mentioned above, it is reasonable to speculate that our method is still applicable to Merlot Reserve.
> ## Q2: Random performance, and typo in Table 1.
> Thanks for the suggestion! In the revised version, we will fix these typos and add the random performance in Table 1, denoted as "Random," with a result of 50.4%.
> ## Q3: Other works coming out after CLIP.
> We appreciate your valuable suggestion. We conducted experiments using more powerful baselines BLIP[2] and ERNIE-ViL [1]. The experimental results are shown in the Table in Q1, after incorporating our module, we achieved 4.0% and 5.5% improvement, respectively, confirming the efficacy of our plug-and-play mudole. Based on the results of the six tested models mentioned above, we believe that our module can enhance the model's audio-visual physical commonsense reasoning ability. We will include the additional results in the modified paper.
> ## Q4: A more concrete example of the motivation of physical knowledge relationship.
> We take a piece of data from PACS as an example:
>
> Question: Which object could be shattered or bent if the other was to strike it forcefully?
>
> Object 1: A large, thick glass jar with a glassy appearance, making a glass-like sound when struck.
>
> Object 2: A small-sized metal shell cellphone, thin in profile with a metallic sheen, producing a metallic sound upon friction.
>
> Our model reasoning for such an example using the following approach:
>
> 1. We aim for the physical knowledge relationships to serve as guidance for the similarity between objects. Through the module, even though object2 may not exhibit the attribute of 'hardness', the shared attribute of hardness among other metallic objects should influence the model's reasoning. For zero-shot objects in the test set (in PACS, all objects in the test set are zero-shot), our approach can leverage attributes from a broader range of objects to aid in reasoning.
>
> 2. For both dynamic and static factors and audio information, we employ the near-neighbor chosen function to extract the Top-5 highest similarity scores from each row, resulting in the affinity matrix A. By utilizing Eq.(6), we derive the final features. Our investigation reveals that in the similarity matrix for dynamic factors, object 1's Top-5 exclusively involve videos depicting striking actions on the object, while object 2's Top-5 encompass actions involving friction. In the static factor, object 1's Top-5 include 2 objects characterized by large volumes and 3 ones possessing a glassy texture, while object 2's Top-5 consist of small metallic sheets. Concerning audio, our exploration demonstrates that object 1's Top-5 emit glass-like sounds, while object 2's Top-5 produce sounds akin to sanding. In the process of reasoning for the question, the key lies in identifying object 1's attributes of 'large volume', and 'fragility', whereas object 2 possesses a 'small volume' and 'hardness'. Therefore, the final reasoning yields the result that object 1 is the answer. Our method implicitly represents these features and identifies objects similar to object 1 and object 2 in dynamic, static, and audio aspects, aiding the model in reasoning about the given problem.
>
> 3. The reasoning process of our proposed method aligns with the findings of Piloto's work [4], which states that human cognition involves the continuous induction of physical concepts, where specific physical concepts are learned through the similarities in physical knowledge across diverse objects. In the PACS dataset, each question is linked to its corresponding physical concept. These concepts encompass both static attributes such as 'color,' 'thickness', as well as dynamic attributes like 'stretchiness', and 'softness'. Within these physical concepts, objects exhibit distinct attributes. Objects sharing similar physical attributes often yield analogous answers to a given question. Our method capitalizes on this characteristic to assist the model in reasoning.
>
> [1] Ernie-vil: Knowledge enhanced vision-language representations through scene graphs. AAAI 2021
>
> [2] Bootstrapping Language-Image Pre-training for Unified Vision-Language Understanding and Generation. ICML 2022
>
> [3] From Recognition to Cognition: Visual Commonsense Reasoning. CVPR 2018
>
> [4] Intuitive physics learning in a deep-learning model inspired by developmental psychology. Nature human behaviour 2022

---

> ### Author Response · Authors · 2023-08-19
>
> Thanks again for your insightful suggestions and comments. Please let us know if our responses have addressed the issues raised in your reviews. We hope that our clarifications and additional results address your concerns and convince you of the merits of our work. We are happy to provide any additional clarifications or experiments that you may need.
>
> Thank you for your time again!
>
> Best,
> Authors

---

### Author Rebuttal · Authors · 2023-08-10

# Our rebuttal material

---

### Decision · Program_Chairs · 2023-09-21

**Decision:**

Accept (poster)

**Comment:**

ive experts reviewed this paper with 4 accepted recommendations and 1 bordering rejection. The area chairs feel that this work makes a very important contribution by introducing a new Physical Audiovisual Commonsense Reasoning dataset. The reviewers did raise some valuable concerns that should be addressed in the final camera-ready version of the paper. The authors are encouraged to make the necessary changes.

It would be also nice if the authors could include some discussions on related work on embodied audio-visual AI tasks (e.g. [1, 2, 3, 4] ) in the final version. These works can also be seen as an inverse physics problem.

[1] Audio-Visual Floorplan Reconstruction. ICCV 2021

[2] Semantic Audio-Visual Navigation. CVPR 2021.

[3] Finding Fallen Objects via Asynchronous Audio-Visual Integration. CVPR 2023

[4] Look, listen, and act: Towards audio-visual embodied navigation. ICRA 2020